# Detection of Hepatitis C Virus Infection from Patient Sera in Cell Culture Using Semi-Automated Image Analysis

**DOI:** 10.3390/v16121871

**Published:** 2024-11-30

**Authors:** Noemi Schäfer, Paul Rothhaar, Christian Heuss, Christoph Neumann-Haefelin, Robert Thimme, Julia Dietz, Christoph Sarrazin, Paul Schnitzler, Uta Merle, Sofía Pérez-del-Pulgar, Vibor Laketa, Volker Lohmann

**Affiliations:** 1Department of Infectious Diseases, Molecular Virology, Section Virus-Host Interactions, Heidelberg University, 69120 Heidelberg, Germany; 2Department of Medicine II, Medical Center, Faculty of Medicine, University of Freiburg, 79110 Freiburg, Germany; 3Department of Gastroenterology and Hepatology, University Hospital Cologne, Faculty of Medicine, University of Cologne, 50937 Cologne, Germany; 4Department of Internal Medicine 1, University Hospital, Goethe University, 60596 Frankfurt, Germany; 5German Center for Infection Research (DZIF), Partner Site Frankfurt, 60596 Frankfurt, Germany; 6Medizinische Klinik 2, St. Josefs-Hospital, 65189 Wiesbaden, Germany; 7Department of Infectious Diseases, Virology, University Hospital Heidelberg, 69120 Heidelberg, Germany; 8Department of Internal Medicine IV, University Hospital Heidelberg, 69120 Heidelberg, Germany; uta.merle@med.uni-heidelberg.de; 9Liver Unit, Hospital Clínic, IDIBAPS and CIBEREHD, University of Barcelona, 08036 Barcelona, Spain; sofiapp@recerca.clinic.cat; 10German Center for Infection Research (DZIF), Partner Site Heidelberg, 69120 Heidelberg, Germany

**Keywords:** HCV, patient sera, cell culture, image analysis, infection, machine learning

## Abstract

The study of hepatitis C virus (HCV) replication in cell culture is mainly based on cloned viral isolates requiring adaptation for efficient replication in Huh7 hepatoma cells. The analysis of wild-type (WT) isolates was enabled by the expression of SEC14L2 and by inhibitors targeting deleterious host factors. Here, we aimed to optimize cell culture models to allow infection with HCV from patient sera. We used Huh7-Lunet cells ectopically expressing SEC14L2, CD81, and a GFP reporter with nuclear translocation upon cleavage by the HCV protease to study HCV replication, combined with a drug-based regimen for stimulation of non-modified wild-type isolates. RT-qPCR-based quantification of HCV infections using patient sera suffered from a high background in the daclatasvir-treated controls. We therefore established an automated image analysis pipeline based on imaging of whole wells and iterative training of a machine learning tool, using nuclear GFP localization as a readout for HCV infection. Upon visual validation of hits assigned by the automated image analysis, the method revealed no background in daclatasvir-treated samples. Thereby, infection events were found for 15 of 34 high titer HCV genotype (gt) 1b sera, revealing a significant correlation between serum titer and successful infection. We further show that transfection of viral RNA extracted from sera can be used in this model as well, albeit with so far limited efficiency. Overall, we generated a robust serum infection assay for gt1b isolates using semi-automated image analysis, which was superior to conventional RT-qPCR-based quantification of viral genomes.

## 1. Introduction

Around 58 million people are chronically infected with HCV worldwide, leading to approximately 300,000 deaths per year (WHO, 2021). HCV is a positive single-strand, enveloped RNA virus from the family of *Flaviviridae* with a genome length of around 9600 nucleotides. Upon infection with HCV, the virus replicates in human liver cells, leading to a chronic infection in around 60 to 85% of cases [1], resulting often in severe liver disease, including cirrhosis and hepatocellular carcinoma (HCC) [2,3]. The high error rate of the viral polymerase leads to a huge genetic heterogeneity of HCV genomes, which are now grouped into eight HCV genotypes (gt) with up to 30% sequence difference and numerous subtypes [4,5,6]. Even within one single patient, HCV exhibits a quasi-species nature with many heterogeneous viral genomes found in the same person [7], which is a major hindrance to the development of a protective vaccine. The introduction of direct-acting antivirals (DAAs) from 2011 onward revolutionized HCV treatment, meanwhile reaching sustained virologic response rates (SVRs) far above 90% [8]. So far, the emergence of resistance-associated substitutions (RAS) against DAAs has been of limited impact (reviewed in [9,10]), but might gain importance in the future with the emergence of new virus strains and subtypes [11]. In the long run, a prophylactic vaccine against the virus is the ultimate goal to reach, being the most effective and least cost-intensive tool to globally extinguish HCV (reviewed in [9]).

The HCV subgenomic replicon system was the first HCV cell culture model, allowing studies on the virus in a laboratory context [12]. It was later discovered that the successful replication of this isolate named Con1 was based on conserved acquired mutations, which were not present in the original sequence [13]. Nowadays, a wide range of reporter replicons, as well as cell-culture-adapted full-length strains with the capability of producing infectious particles, is available for many different genotypes, but all based on cloned viral isolates [14,15,16,17,18,19,20]. However, until now, culturing WT HCV virus from patient serum remains difficult, thereby hampering studies on the natural heterogeneity of HCV quasi-species and their functional implications.

Progress in the field of HCV WT replication in cell culture has been made due to essential findings concerning two cellular host factors, namely, Phosphatidylinositol 4-kinase IIIα (PI4KA) and SEC14L2. PI4KA is a cellular lipid kinase present in primary hepatocytes as well as in the Huh7 cell lines mainly used for culturing HCV. However, in contrast to healthy hepatocytes, PI4KA is overexpressed in hepatoma cell lines [21,22]. It could be shown that a combined, limited chemical inhibition of PI4KA and Casein kinase I alpha (CKIα) [23] phenocopied the effect of distinct cell culture adaptive mutations in the viral proteins NS5A and NS5B [21]. This so-called PCi treatment thereby facilitated replication of HCV gt1b WT sequences in Huh7 cells [24]. SEC14L2 is a cytosolic lipid-exchange protein, which is expressed in primary human hepatocytes, but not in Huh7 cells. It was discovered as a cellular host factor supporting the replication of HCV WT pan-genotypically in a screen using Huh7.5 cells expressing a cDNA-library [25]. However, overexpression of SEC14L2 in Huh7 cells acts most efficiently on gt1b and has variable effects on replication levels of other gts [26].

Replication of HCV WT isolates from patient serum using either PCi treatment or SEC14L2 overexpression could be achieved for a small number of sera, mainly with high titers obtained from patients after liver transplantation [21,25,26]. We further found that a combination of PCi treatment and SEC14L2 overexpression resulted in additive effects in enhancing the replication of two gt1b WT replicons in Huh7 Lunet cells [24], but potential benefits in culturing serum-derived virus remained unexplored.

Here, we aimed to establish a cell culture model for studying the replication of serum-derived HCV. We generated a robust infection assay for gt1b isolates based on cells expressing SEC14L2 and PCi treatment. The detection of HCV-protease-dependent nuclear localization of GFP, analyzed by whole-well imaging and automated image analysis with visual validation of positive cells, proved to be superior to conventional RT-qPCR-based quantification of viral genomes.

## 2. Materials and Methods

### 2.1. Cell Culture

Huh7 Lunet CD81H/MGN/SEC14L2 (Huh7 cell line overexpressing the entry factor CD81, SEC14L2, and additionally harboring a MAVS-GFP-NLS-reporter system with blasticidin, puromycin, and neomycin as selection markers) have been described previously [24]; Huh7 LucUbiNeo Con1 ET [27]); and Huh7 LucUbiNeo JFH [28] have been described previously. Huh7 LucUbiNeo GLT1 cells were generated by electroporation of in vitro transcribed RNA generated from plasmid pFK i389 LucUbiNeo NS3-3′ GLT1 WT into Huh7-Lunet cells stably expressing SEC14L2, followed by selection with 500 µg/mL G418.

Cells were cultured in complete Dulbecco’s modified Eagle’s medium (DMEM) supplemented with 2 mM L-glutamine, 0.1 mM nonessential amino acids, 100 U/mL of penicillin, 100 mg/mL of streptomycin (all Gibco), and 10% fetal calf serum (FCS) at 37 °C and 5% (*v*/*v*) CO_2_. Medium of cells harboring selectable replicons was supplemented with 500 µg/mL G418.

Huh7-Lunet CD81H/MGN/SEC14L2 (Huh7-Lunet C/M/S) cells were cultured in the presence of G418 (1 mg/mL), puromycin (2 µg/mL), and blasticidin (5 µg/mL) to maintain transgene expression in all cells.

#### Plasmids

For replication studies, gt1b subgenomic replicons (SGRs) with a luciferase reporter were used and have been described previously: pFK i341 PiLuc NS3-3′ Con1, pFK i341 PiLuc NS3-3′ Con1 ΔGDD (Replication deficient negative control) [12,13]); and pFK i341 PiLuc NS3-3′ GLT1 [24]. To generate cell lines stably harboring the GLT1 SGR, pFK i389 LucUbiNeo NS3-3′ GLT1 WT was generated by replacing the NS3-3′ region of the respective Con1 replicon [29] by the GLT1 sequence. For production of infectious virus, constructs harboring either the full viral sequence from one isolate (genotype 3a: pUC DBN WT [19]; genotype 1a pUC TN WT [15]) or for genotype 2a, a chimera with the structural proteins up to the C3 junction in NS2 from the J6 isolate and the remainder of the polyprotein from the highly replication fit JFH1 isolate (HCVcc) [18].

### 2.2. Patient Material and Ethics Statement

All patient sera used in this study were obtained from Julia Dietz and Christoph Sarrazin (German Center for Infection Research (DZIF), External Partner Site, Frankfurt, Germany), Uta Merle (University Hospital Heidelberg, Department of Gastroenterology, Heidelberg, Germany), Christoph Neumann-Haefelin and Robert Thimme (University of Freiburg, Department of Medicine, Freiburg, Germany), Sofía Pérez-del-Pulgar [30] (University of Barcelona, Liver Unit, Hospital Clínic, IDIBAPS, CIBERehd, Barcelona, Spain) and Paul Schnitzler (University of Heidelberg, Department of Infectious Diseases, Virology, Heidelberg, Germany). The use of the patient sera for the present study was approved by the Ethics Committee of the Medical Faculty of Heidelberg University (S-677/2020).

### 2.3. Serum Infection

Huh7 Lunet C/M/S cells were seeded at a density of 2 or 1.5 × 10^4^ cells/well in 24-well plates the day before infection. Approximately 24 h later, cells were inoculated with 80 µL patient serum mixed with 320 µL serum-free medium, and spin incubation was performed for 2 h at 1000 g at RT. JC1cc administered with an MOI of 1 served as positive control. Samples were left at 37 °C for 2 to 3 h; next, the serum was removed and cells were incubated using complete DMEM supplemented with PI4KA/CKIα inhibitor treatment (PCi). According to the specific experiment, additional drugs (Daclatasvir (DCV), Legalon SIL (SIL), and Alisporivir (Alis)) were administered. In parallel, RNA from 200 µL serum was extracted using the NucleoSpin Virus kit (Macherey-Nagel, Düren, Germany) and analyzed via RT-qPCR, to determine the viral load in copy number/mL of the patient serum. The samples were incubated for 72 h and harvested according to the respective readout, which consisted of either RT-qPCR or the newly established image-based analysis. For RT-qPCR, RNA extraction was performed using the NucleoSpin RNA Plus kit (Macherey Nagel, Düren, Germany), and the samples were then further processed using the BIO-RAD CFX Opus 96 Real-Time PCR System Thermal Cycler.

### 2.4. Automated Image Analysis (AIA)

For the image-based analysis, cells were fixed with 4% formaldehyde in PBS. After incubation at RT for 20 min, cells were washed 3 times with PBS. Next, cells were either stained with 4′,6-Diamidin-2-phenylindol (DAPI, Invitrogen, Darmstadt, Germany) for 1 min at a dilution of 1:4000 or stained with Hoechst 33342 (Sigma, Darmstadt, Germany) for 5 min at a dilution of 1:2000 and then washed 3 times for 5 min. Images of the whole plates were taken using a Zeiss Cell Discoverer 7 microscope (CD7; Zeiss, Oberkochen, Germany). Per condition, three wells were used containing approximately 3 × 10^5^ cells in total. Afterwards, the images were analyzed via ilastik, an interactive learning and segmentation toolkit to detect co-localization of the GFP and DAPI signal and hereby find and mark infected cells. To segment nuclei in stitched images, we trained a Random Forest classifier based on the DAPI signal using ilastik [31], which predicts semantic class attribution (nucleus or background) for every pixel. Objects obtained in this way were subsequently filtered by size to exclude unusually small or large nuclei, which often represent aberrant biological structures or microscopy artifacts. The Hysteresis algorithm was used to separate nuclei in close proximity to each other. In the next step, using ilastik object classification workflow, a machine learning algorithm was trained to classify objects into two categories—segmenting and classifying cells based on differences in GFP localization patterns between infected and non-infected cells (ER localization = non-infected, nuclear localization = infected).

The training set of data was arbitrarily selected, and the same machine learning algorithm was used for the pixel and object classification in all images. Three training rounds were performed to reduce the background of false-positive hits.

Cells marked as positive by the program were evaluated visually to reduce the false-positive background to zero.

### 2.5. Pharmacological Inhibitors and Drugs

The specific PI4KA inhibitor PI4KA-F1 was provided by Glaxo-Smith-Kline [32]; the CKIα-specific inhibitor H479 was a generous gift from Raffaele de Francesco (Milan, Italy) and has been described previously [23]. Legalon SIL was obtained from Madaus (Troisdorf, Germany), and Alisporivir from Debiopharm (Lausanne, Switzerland). The NS5A inhibitor Daclatasvir was obtained from Bristol Myers Squibb (New York, NY, USA).

### 2.6. Primers and Probes for RT-qPCR

S_GAPDH (GAAGGTGAAGGTCGGAGTC); A_GAPDH (GAAGATGGTGATGGGATTTC); A_219_JFH (GGGCATAGAGTGGGTTTATCCA); S_146_JFH (TCTGCGGAACCGGTGAGTA); A165_Con1 (TACTCACCGGTTCCGCAGA); S59_Con1 (TGTCTTCACGCAGAAAGCGTCTAG).

GAPDH (6-VIC-CAAGCTTCCCGTTCTCAGCCT-TAMRA); CON1 (6-FAM-TCCTGGAGGCTGCACGACACTCAT-TAMRA); JFH (6-FAM-AAAGGACCCAGTCTTCCCGGCAATT-TAMRA).

### 2.7. Production of Viral Particles

To produce JC1cc infectious particles, electroporation of in vitro transcribed JC1cc RNA was performed and cells were incubated for 3 days. The medium was then filtered through a 0.45 µm filter and stored at 4 °C overnight. The cells were supplied with fresh DMEM. The next day, the DMEM was again removed and filtered. Filtered supernatant was concentrated using a Centricon Plus-70 centrifugal filter device (100 K nominal molecular weight limit; Merck, Darmstadt, Germany) according to the manufacturer’s instructions. Infectivity of the concentrated supernatant was determined via Tissue Culture Infectious Dose 50 (TCID50) as previously described [24] with the calculation based on the method of Spearman and Kärber [33].

### 2.8. Cell Viability Assay

Cell viability was measured using the Cell Titer Glo Luminescent Cell Viability Assay (Promega, Madison, WI, USA). Cells were seeded in 96-well plates with 1 × 10^4^ cells/well and incubated in complete DMEM supplemented with the various drug concentrations in triplicates. Three wells were left empty for background control. After 3 days, the samples were measured according to the manufacturer’s instructions using the Mithras LB 940 plate reader (Berthold Technologies, Bad Wildbad, Germany). The results were analyzed using Excel.

### 2.9. In Vitro Transcription, Electroporation, and Luciferase Activity Assay

In vitro transcription, electroporation, and luciferase activity assay were performed as described previously [13]. Briefly, 10 µg plasmid DNA was linearized with SpeI (Con1- and GLT1-based plasmids), MluI (JC1cc), and XbaI (TN WT and DBN WT) and used as input for the in vitro transcription reaction. Then, 2.5–5 µg purified HCV in vitro transcripts or the respective HCV RNA copy numbers (CN) of patient-derived RNA was electroporated into 2–4 × 10^6^ Huh7 Lunet C/M/S cells. Four hours after seeding the electroporated cells in 12 well dishes, drug treatments were performed. For luciferase activity assays, cells were lysed at the respective time points and luciferase activity was measured using a Lumat LB9507 tube luminometer (Berthold Technologies). To account for variances in input RNA abundance, measured values were normalized to the luciferase activity at 4 h.

### 2.10. RT-qPCR

RT-qPCR was performed measuring GAPDH-RNA as a control gene expressed ubiquitously in the host cells. A second primer set was used to detect HCV-RNA and was designed according to the HCV genotype of the specific experiment. As a first step, a mastermix was prepared containing the two primer sets, two probes detecting GAPDH and HCV, the qScript XLT 1-step RT-PCT ToughMix (Quantabio, Beverly, MA, USA), and distilled water to add up to the final volume. The mastermix was then pipetted into a 96-well PCR plate (hard-shell, thin-well, BioRad, Feldkirchen, Germany) using 12 µL per well. Next, a standard adjusted to the respective HCV genotype of the experiment was given to the first 24 wells ranging from 1 × 10^9^ to 1 × 10^2^ HCV-transcripts/µL. The samples for analysis (prepared from cells using the NucleoSpin RNA Plus kit by Macherey-Nagel, Düren, Germany) were pipetted into the remaining wells. 3 µL of standard/sample was used per well and measurements were performed in triplicates. After preparation, the plate was sealed and analyzed with the BIO-RAD CFX Opus 96 Real-Time PCR System Thermal Cycler using the “Taqman Quanta program” (Step 1: 50 °C for 10 min; Step 2: 95 °C for 1 min; Step 3: 95 °C for 10 secs; Step 4: 60 °C for 1 min; Repeat from Step 3 for 39 times). Results were exported to an Excel file and analyzed there. If not stated otherwise, experiments were performed with three technical replicates per biological replicate.

### 2.11. Electroporation of HCV RNA from Patient Serum

Electroporation was performed as previously described [13]. In brief, viral RNA extracted from patient serum was electroporated into 2 × 10^6^ Huh7 Lunet C/M/S cells using a GenePulser system (BioRad, Feldkirchen, Germany) at 975 µF and 166 V in a cuvette with a gap width of 0.2 cm. Cells were then diluted in 8 mL DMEM and seeded into 12-well plates using 500 µL cell suspension and 500 µL fresh DMEM per well. PCi treatment was added to all samples, and as a negative control, half of the wells were additionally treated with DCV. Samples were harvested after 72 h and analyzed by RT-qPCR or by automated image analysis.

### 2.12. Liposome-Based Transfection

The day before transfection, cells were seeded in 6-well plates using 2 to 2.5 × 10^5^ cells per well, leading to 80 to 90% confluence the next day. On the day of transfection, 3.75 µL of Lipofectamine MessengerMax (Thermo Fisher, Bleiswijk, Netherlands) reagent was mixed with 125 µL OptiMEM (Gibco, Carlsbad, CA, USA) and incubated for 15 min at RT. The RNA for transfection was given into 125 µL OptiMEM as well; the two tubes were combined and then incubated for 5 min. The 250 µL end product was pipetted into the 6-well plate, which had been preincubated with OptiMEM at 37 °C for approximately 30 min. After 4 h, OptiMEM was removed and substituted by complete DMEM. The next day, the cells were split according to the respective experiment.

### 2.13. Ultracentrifugation

To prepare a sucrose cushion, 20% sucrose was dissolved in TNE buffer (12 mM Tris-HCl (pH 8.0), 120 mM NaCl, 2 mM EDTA in dH_2_O), filtered sterilely, and stored at 5 °C. To concentrate viral RNA from patient serum, 2.5 mL sucrose in TNE buffer was overlaid with 10 mL patient serum mixed with 20 mL PBS in an ultracentrifugation tube. Ultracentrifugation was performed at 28,000 rounds per minute (rpm) and 4 °C for 2 h using a Beckmann rotor SW32. Afterward, centrifuge tubes were emptied and dried, leaving a pellet containing the viral particles. The pellet was dissolved in 750 µL TRIzol (Ambion, Carlsbad, CA, USA), and RNA extraction was performed according to the standard protocol.

### 2.14. TRIzol LS Extraction

For TRIzol LS extraction, 200 µL of serum was mixed with 600 µL TRIzol LS reagent and incubated for 5 min at RT. Then, 160 µL chloroform was added, and the mixture was shaken vigorously by hand for 15 s. After incubation at RT for 2–3 min, centrifugation at 12,000× *g* for 15 min at 4 °C was performed to achieve phase separation. The upper phase was transferred into a new tube, mixed with 400 µL isopropanol, and incubated at RT for 10 min. Next, the mixture was centrifuged at 12,000× *g* for 10 min at 4 °C, and afterwards, the supernatant was removed, leaving the RNA pellet. Then, 1 mL 75% ethanol was added, and the sample was centrifuged at 7500× *g* for 5 min at 4 °C. The ethanol was removed, the pellet was air-dried for 5 min and afterward resuspended in 30 µL dH_2_O. The sample was incubated at 55–60 °C on a heat block, and then stored at −80 °C.

### 2.15. Statistical Analysis

Statistical analysis was performed in GraphPad Prism (Version 8.4.3) using the unpaired Student’s *t*-test.

## 3. Results

### 3.1. Establishment of Optimal Conditions for Serum Infection

In this study, we aimed to establish a cell culture model for robust detection of serum-derived HCV infection. To this end, we used Huh7-Lunet CD81H/MAVS-GFP-NLS/SEC14L2 cells (Huh7-Lunet C/M/S) stably expressing three transgenes: CD81, a crucial entry receptor, which is not efficiently expressed in Huh7-Lunet cells [34]; SEC14L2, facilitating RNA replication of HCV WT isolates [25]; and GFP fused to a nuclear localization signal (GFP-NLS) and the C-terminus of MAVS including the NS3-4A cleavage site, allowing the detection of HCV infection by nuclear transfer of GFP-NLS [21,35]. We had previously identified a SEC14L2 independent strategy to promote RNA replication of HCV gt1b WT isolates based on combined inhibition of PI4KA/CKIα (PCi treatment) [21]. Since we found some additive effect upon a combination of PCi and SEC14L2 [24], we first aimed to optimize PCi concentrations in Huh7-Lunet C/M/S, using subgenomic reporter replicons. We used two gt1b WT genomes with variable replication capacities, Con1 and GLT1 [24], since this genotype has been shown to be most efficiently stimulated by SEC14L2 [25,26] and PCi treatment [21]. Using a fixed concentration of 5 µM of the CKIα-inhibitor H479, we tested variable concentrations of the PI4KA-inhibitor F1 (Figure 1A). A moderate enhancement of replication for both isolates could be detected up to a concentration of 10 nM followed by a decline at higher concentrations, similar to our previous results [21]. The combination of 5 µM H479 and 5 nM F1 was chosen for all later experiments.

We next titrated the NS5A-inhibitor Daclatasvir (DCV), which we planned to use as a control to define active replication in the infection assay. Already at 0.1 nM, replication of both isolates was suppressed to the level of a replication-deficient control (ΔGDD, Figure 1B). Since no cytotoxicity was observed at any concentration (Figure 1C), we chose 1 nM DCV for further experiments, and to inhibit WT isolates with potentially lower DCV sensitivity.

To explore the dynamic range of various detection methods in greater detail, we included two additional antiviral agents, namely, Legalon SIL (SIL) and Alisporivir (Alis), both being potential candidates for patients failing to be cured by current DAA regimens [36]. Legalon SIL exhibited antiviral effects against HCV in some patients [37], with a yet poorly understood selectivity toward certain isolates and genotypes [38]. Alisporivir is a non-immunosuppressive Cyclophilin A inhibitor efficiently blocking the interaction with NS5A [39,40,41]. Both compounds were tested in Huh7 Lunet C/M/S cells with the same experimental setup as used for DCV. The Con1 WT replicon was entirely inhibited by SIL concentrations above 250 µM (Figure 1D). For GLT1, a drop in replication was observed as well, but levels stayed clearly above the negative control hinting at a partial resistance, as observed before for other HCV isolates like JFH1 [38]. Due to inhibitory effects on cell growth upon rising concentrations (Figure 1E), 250 µM was chosen as a standard concentration. Also, in the case of Alisporivir, GLT1 was less sensitive than Con1, requiring 1 µM Alisporivir for complete inhibition (Figure 1F). This concentration was used in subsequent experiments since it further lacked cytotoxic effects (Figure 1G).

All data generated so far were based on luciferase activity measurement upon transfection of reporter replicons. However, RT-qPCR was one method of choice to measure HCV replication upon infection with HCV-positive sera. We therefore aimed to define the dynamic range of the two assays, using DCV, Legalon-SIL, and Alisporivir. We chose cell lines harboring persistent, selectable reporter replicons, since electroporation of in vitro transcribed viral genomes would have generated a high background in RT-qPCR. To that end, we used already-established cell lines based on Con1 [27], JFH1 [28] and a newly generated cell line containing GLT1 WT. All three cell lines were treated with the previously determined standard concentrations of SIL, Alisporivir, or DCV, harvested after 24, 48, and 72 h and analyzed for luciferase activity (Figure 2A, upper panels) or by RT-qPCR (Figure 2A, lower panels). In principle, we observed the same pattern of sensitivity to the different drugs in both assays for all three replicons, with Con1 being highly sensitive to all compounds, whereas GLT1 was less sensitive and JFH1 was completely resistant to Legalon-SIL. However, while the reporter assay offered a reduction upon full inhibition of more than two logs, the dynamic window of the RT-qPCR assay was much lower, as expected [28]. Thorough statistical analysis was precluded by the low number of repetitions for both assays (n = 2).

Finally, we validated our model in infection, using JC1cc and GLT1 patient serum as positive controls; the latter had already shown robust cell culture replication upon PCi treatment in a previous study [21] (Figure 2B). JC1cc RNA levels were reduced by more than two logs upon Alisporivir and DCV treatment and about 3-fold by Legalon-SIL, comparable to the results obtained with the replicon cells (Figure 2A, right panels). In the case of GLT1 serum infection, no pronounced additive effect of the combination of PCi treatment and SEC14L2 overexpression in comparison to PCi treatment alone could be detected when comparing our results to the results of a previous study [21]. This was also in line with previous observations using the GLT1 replicon ([24] and Figure 1A). The drug treatments clearly reduced HCV RNA levels, indicating a successful infection followed by viral replication. However, the reduction compared to the untreated control was only fivefold. The much smaller window between infection and DCV-treated samples in comparison to the JC1cc control could be due to the difference in replication efficiency in cell culture, which is exceptionally high for JC1cc. It could further point to a larger impact of the remaining input RNA when working with patient sera, e.g., due to a higher number of defective particles attaching to, but not entering, the cells. However, the level of reduction detected by RT-qPCR did not reach statistical significance. Of note, the GLT1 patient serum appeared to be more sensitive to SIL treatment than the GLT1 WT subgenomic replicon, and also for JC1cc, a slightly more pronounced inhibitory effect was observed. This could be explained by SIL acting on multiple steps of the HCV life cycle [42,43], including steps like cell entry, which are relevant during serum infection but not in subgenomic replicons [44].

Overall, we optimized a cell culture model based on Huh7-Lunet C/M/S cells by combining SEC14L2 expression and PCi treatment, principally allowing the analysis of HCV infection using patient sera. However, the dynamic range obtained by RT-qPCR was relatively low, even in the case of a high-titer post-transplant serum.

### 3.2. Establishment of a Quantitative, Semi-Automated Image Analysis Pipeline for Serum Infection

We then started testing the first cohort of 27 HCV-positive patient sera, mainly gt1b, based on a previously established serum infection protocol [21] (Figure 3A, Appendix A). DCV treatment was included to define bona fide viral replication from viral input RNA. We additionally determined the viral RNA titer of all sera, using the same RT-qPCR assay as for the quantification of viral replication after infection (Appendix A). Unfortunately, HCV RNA was clearly detectable post-infection (Ct < 36) only for the two sera with the highest titers (Appendix A), and no infection showed a significant reduction upon DCV treatment (too complex to show). We therefore decided to establish an image-based readout utilizing the GFP reporter system contained in the Huh7-Lunet C/M/S cells [35]. Due to our previous experience, suggesting an overall very low number of infection events [21], we aimed at imaging whole wells and implementing an automated readout to generate accurate and reproducible results. To that end, a Zeiss Cell Discoverer 7 microscope (CD7) was used combined with automated image analysis by ilastik [31], a machine learning image analysis tool, segmenting and classifying cells based on differences in GFP localization patterns between infected and non-infected cells (ER localization = non-infected, nuclear localization = infected) (Figure 3B). Ilastik was trained with images of cells infected with JC1cc using DCV treatment and non-infected cells as negative controls, resulting in a robust, significant detection of infected cells (Figure 3C). After three iterative training sets, the background of false-positive detections could be reduced from 2% to 0.01–0.02% (Figure 3D), corresponding to about 100 cells in a 24-well dish. However, these low background levels still surpassed the true-positive hits, so automated image analysis (AIA) had to be followed by visual validation of the annotated positive hits. Nevertheless, this still was a major gain in efficiency, since only 100 instead of ~300,000 to 500,000 cells had to be visually inspected. Visual validation of hits automatically assigned by ilastik revealed not a single truly positive cell in any of the JC1cc infected samples with DCV treatment or in the negative controls, arguing for the stringency of this analysis pipeline. The AIA followed by a visual validation thereby allowed the quantification of infected cells in the absence of any false-positive signals (Figure 3E).

### 3.3. Semi-Automated Image Analysis Revealed More Consistent Results upon Serum Infection than RT-qPCR

We next aimed to analyze whether semi-automated image analysis enabled a thorough determination and quantification of infections using HCV WT isolates from serum. Based on the previous experience, samples with high viral loads of >10^6^ RNA copies per ml were chosen for further infection experiments to compare RT-qPCR and image analysis (Table 1). Then, 11 out of 29 gt1b sera were taken after liver transplantation, including GLT1, providing a better chance of success due to high viral loads and the potential lack of neutralizing antibodies. The limited amount of available sera (1 mL or below), allowed for triplicate infections for both methods, including the DCV controls.

HCV RNA was detectable by RT-qPCR in 11 infections; however, a statistically significant drop in RNA copy numbers upon DCV treatment was only found in one case (serum 81), and more than twofold reductions lacking significance in an additional two cases (GLT1 and serum 2130, Figure 4A, Table 1). The fully automated image-based analysis revealed a slightly varying background of 0.02–0.07%, judged by the DCV controls, due to disruptive factors like locally increased cell density or ill-focused images. Statistically significant reductions upon DCV treatment were obtained only for three sera, 60, 84, and GLT1 (Figure 4B), suggesting that the number of positive cells was below the background of false-positive detections in most cases. In contrast, the automated analysis combined with visual validation identified infected cells with verified bona fide nuclear GFP signal in overall 11 cases, ranging from a total number of 1 to about 200 infected cells per well (Figure 4C, Table 1). All sera showing clearly detectable or significant differences in the RT-qPCR analysis (Figure 4A) or the automated image analysis (Figure 4B), respectively, indeed were verified to contain infected cells. In the case of serum 14 and 5651, only a single infected cell was detected in one or two wells, respectively. Importantly, again not a single positive cell with nuclear GFP was found in any of the DCV-treated controls. Overall, the automated image analysis combined with visual validation offered a highly sensitive proof of successful infection due to the lack of any background, thereby allowing even a single infected cell to be identified, in contrast to genome detection by RT-qPCR.

We next analyzed whether the titer and the transplant status of the tested patient sera did have an impact on the respective success of a serum infection (Figure 4D). Indeed, we found a significant correlation between serum RNA titer and successful detection of infection events. All sera with titers above 1 × 10^7^ RNA copies/mL resulted in detectable infected cells, but based on the limited data, we could not define a lower threshold. Post-transplant sera had a higher success rate (8/11 gt1b sera compared to 7/23 non-transplant sera), but also had on average higher titers.

Taken together, we were able to establish a robust infection model for HCV patient sera, based on a semi-automated image analysis pipeline using Huh7-Lunet C/M/S cells.

### 3.4. Establishment of Transfection-Based Delivery of Serum-Derived HCV WT RNA

Since titers above 10^6^ RNA copies/mL are rare, and the infectious capacity of the serum samples might have been affected by multiple factors such as neutralizing antibodies, damaged infectious particles, cytotoxicity of some serum components, and many more, we tested whether transfection of viral RNA extracted from serum samples would lead to more effective HCV WT replication in Huh7-Lunet C/M/S cells. Such an approach would further allow upscaling by using a higher volume of serum for RNA extraction. As a first step, a transfection protocol was established comparing lipid-based transfection and electroporation of in vitro transcribed HCV WT RNA (Figure 5). To that end, full-length in vitro transcripts of the GLT1 WT consensus sequence, the gt1a isolate TN WT [15], and the gt3a isolate DBN WT [19] were used in increasing concentrations. Measurable HCV RNA levels in RT-qPCR could reliably be observed starting from an RNA copy number of 10^8^ with both methods, lipid-based transfection and electroporation (Figure 5A,B). This was the case for all three isolates independently of the genotype. However, when compared to the DCV control, only the GLT1 isolate exhibited a reproducible reduction in HCV copy numbers under all conditions, suggesting that gt1a and gt3a WT replication was not supported to the same extent as gt1b upon SEC14L2 expression and PCi treatment, in line with previous studies [21,26]. Interestingly, DCV resulted in a far stronger decrease in the RNA signal after electroporation, suggesting that naked viral input RNA might be degraded more efficiently than in vitro transcripts complexed in transfection reagents, resulting in a lower background.

In the automated image analysis, a rise in positive cell numbers compared to DCV using the GLT1 isolate could be detected starting from 10^9^ RNA copies upon electroporation and 10^10^ RNA copies upon lipid-based transfection. Using the visual validation, a few cells with nuclear GFP signal were already found in samples transfected with 10^7^ RNA copies upon electroporation (Figure 5C) and 10^8^ copies upon lipid-based transfection (Figure 5D). Again, no validated positive cells were found after DCV upon electroporation (data omitted), suggesting that translation of NS3-4A from transfected input RNA was not sufficient to confer nuclear GFP translocation. In contrast, a high number of positive cells were detected in the DCV-treated control after lipid-based transfection, starting from 10^9^ RNA copies, because DCV was added only 24 h after the addition of the transfection mixture, after splitting of the cells, to avoid interference of both treatments. In line with previous studies, this was obviously too late to impede nuclear GFP translocation [24,35,45]. TN WT and DBN3a WT did not show a significant rise in positive cells with both transfection methods, even when high amounts of input RNA were used. Upon visual validation, a few infected cells were detected with an input of 10^10^ RNA copies for both isolates and transfection methods. Therefore, also non-gt1b isolates appeared to be able to replicate in this assay, although to a much lower extent. The combination of SEC14L2 expression levels achieved in these cells combined with PCi treatment might not have been optimal for gt1a and gt3a isolates and might require further optimization.

Overall, electroporation was chosen for further studies, since positive cells could already be detected at lower input RNA copy numbers and due to lower background in RT-qPCR.

We used 10 patient sera to compare infection with transfection of viral RNA (Figure 6). For four sera, RNA was extracted from 1 mL volume using TRIzol LS, and for six serum samples, we first aimed to concentrate the virus from approximately 10 mL serum by ultracentrifugation prior to RNA extraction. However, ultracentrifugation from larger volumes did not necessarily result in increased viral RNA yields, indicating that a substantial amount of viral RNA was lost (Table 2). Transfection and infection were compared by RT-qPCR and by image analysis.

For transfection of RNA derived from serum 2130, we obtained a significant increase in viral RNA and a substantial number of positive cells upon AIA with visual validation (Figure 6A,C), providing proof of concept that transfection of serum-derived viral RNA indeed is a feasible approach to study replication of WT isolates in cell culture. However, no significance was reached with AIA, due to variable seeding densities after electroporation, resulting in a high background (Figure 6B), and only two additional transfected RNAs yielded single positive cells (Figure 6C, left panel). The same set of samples was positive upon infection plus an additional four sera, which were negative upon transfection (Figure 6C, right panel). The reason for the failure of RNA extracted from GLT1 to mount replication upon transfection despite efficient RNA extraction was not clear.

Overall, transfection of viral RNA extracted from serum so far provided no benefit compared to infection, albeit possible in principle. A thorough optimization of enrichment of viral RNA from larger serum volumes still holds the promise to widen the scope of sera applicable for testing in cell culture.

## 4. Discussion

In this study, sera mostly from gt1b HCV-infected patients were tested on their capacity to establish successful RNA replication in Huh7-Lunet C/M/S cells. A newly established quantitative image-based analysis with visual validation proved to be virtually free of any background, allowing a robust detection even of single infection events.

In this study, RT-qPCR failed to generate statistically significant differences compared to DCV treatment, which is an essential control, likely due to the high background of cell-associated virions. This view is supported by the reduced background observed upon electroporation of purified viral RNA. A potential reason for the high background might be the spin infection, which was part of our protocol [21], but not used in other studies [25,26]. We therefore used an alternative detection method based on viral protease activity [35]. This system is well established in the context of HCVcc infection [46,47], also in combination with automated image analysis [24,35] and in a proof-of-concept to detect infection of serum-borne viruses [21]. Due to the expectedly low number of infection events, imaging of whole wells combined with automated image analysis was an essential component for a reproducible analysis pipeline for serum infection. In contrast to the RT-qPCR, background arose from false-positive assignments of nuclear GFP signals, e.g., in areas with high cell density, which could be reduced to 0.02% by iterative training of the software. Importantly, the background could be completely eliminated by visual control, generating a robust assay, allowing the unequivocal detection of even single-infection events with reasonable effort. The background might be further reduced by alternative reporter systems based on split-GFP [48].

Since our focus was on testing rather rare high-titer gt1b sera, we mainly had to rely on historical samples. In total, infection events were detectable in 15 out of 35 sera, with a significant correlation of high titers and successful infection. Still, there was no clear threshold titer, and titers did not correlate at all with the number of infection events. The variability observed was likely due to unknown storage conditions/freeze–thaw cycles [49] and the presence of neutralizing antibodies (reviewed in [50]). Among those sera with known storage history lacking freeze–thaw cycles (Table 2), four out of six gave rise to infections, despite comparably low titers, suggesting that the success rate might be higher in freshly acquired samples. Post-transplant sera also had a higher-than-average success rate (8/11, versus 7/24 non-transplant sera), which could be due to high average titers, lack of neutralizing antibodies, and the presence of variants with higher fitness. Previous studies have shown that HCV reinfection after liver transplantation results in a reduction in HCV genome sequence variety [30,51], which can be associated with increased cell entry competence [52,53]. Interestingly, the GLT1 patient serum showed the highest number of infection events, and has been shown to replicate with far higher efficiency than common gt1b isolates like Con1 [24], pointing toward selection for increased RNA replication fitness after liver transplantation (LTx), with yet unknown sequence determinants.

We focused our analysis on gt1b sera, since previous studies have shown that SEC14L2 expression as well as PCi treatment works most efficiently/exclusively in this context. PCi treatment mimics the mode of action of distinct classes of adaptive mutations [21], but since only gt1b isolates are capable of replicating with a single adaptive mutation, this measure is not sufficient for other genotypes. SEC14L2 acts pan-genotypically in the context of selectable genomes [25], but also here efficiency dramatically differs among genotypes, again with gt1b being most efficiently stimulated in transient replication assays [26]. We combined both measures and carefully optimized them here, due to additive effects observed with reporter replicons [24]. We did not further evaluate the sole use of SEC14L2 as in previous studies [25,26] due to limited amounts of available sera. The modest additive effects observed here upon the addition of PCi treatment were in line with our previous data and might support bulk detection of replication by reporter activity or RT-qPCR. Whether or not the addition of PCi supports microscopy-based detection should be evaluated side-by-side in infection experiments in the future, since access to these compounds could be a major bottleneck for a broader use of the assay. However, we confirmed that even the combined use of PCi and SEC14L2 was very inefficient in enhancing the replication of cloned gt1a and gt3a wt isolates TNwt and DBNwt, respectively [15,19], and further failed to detect infection based on a gt1a post-transplant serum. Still, previous studies succeeded in infecting SEC14L2 expressing cells, based on gt3a sera [25,26]. This might either be due to differences in SEC14L2 expression levels, or due to the presence of specific high replicator isolates in the high-titer post-transplant sera used [25], comparable to our GLT1 isolate [21,24]. A better understanding of determinants of cell culture adaptation in Huh7 cells or the establishment of alternative cell culture models is required to allow robust serum infections beyond gt1b.

So far, we have focused our studies on single-round infection analyses, since we did not expect that infected cells would generate detectable levels of infectious virus. This assumption was based on our experience with isolate GLT1, requiring a year of continuous passaging and the acquisition of >20 mutations to allow efficient virus production and spread [24], which we assume is due to a defect of Huh7 cells in supporting HCV assembly. Still, some serum-derived isolates might be more prone to virus production in this model, and it might therefore be promising to try the passaging of infected cells or supernatants, followed by the sequencing of virus populations prior to and after infection. Such experiments are facilitated by the reporter contained in Huh7-Lunet C/M/S, allowing us to monitor the infection outcome even in living cells.

To overcome the limited availability of high-titer sera and to avoid factors interfering with serum infection like neutralizing antibodies, damaged viral particles, cytotoxic serum components, etc., we further evaluated the possibility of transfecting viral RNA purified from serum. However, already the validation based on in vitro-transcribed viral genomes demonstrated the need for more than 10^7^ viral genomes for the detection of HCV-positive cells, probably due to the accessibility of naked viral RNA to RNases during transfection. Still, the principle feasibility of transfection of purified RNA from serum offers the opportunity for upscaling. We so far failed to enrich viral RNA from larger volumes by ultracentrifugation, probably due to the heterogeneous density of HCV particles in serum [54]. A careful optimization of virus purification and concentration methods, e.g., using aqueous two-phase systems, might enable efficient purification of viral RNA from large serum volumes [55]. Alternatively, the infectivity of sera might be improved by the removal of neutralizing antibodies.

DAAs have revolutionized HCV therapies, offering safe and effective treatment options for almost all patients. However, some viral isolates have been identified with an enhanced amount of clinically relevant RAS, conferring resistance to current drug regimens [56,57,58,59] and new HCV subtypes, mostly deriving from sub-Saharan Africa, and revealed a higher prevalence of clinically relevant RAS, leading to an enhanced rate of treatment failure in these patients [60]. Here, a serum-based infection assay could help to find personalized treatment options including the use of drugs with so far unpredictable response patterns. Legalon-SIL is a prime candidate for such an approach, due to its clinically proven efficacy against HCV, with many patients showing a strong decline in viral load, but some not, in the absence of clear correlations to viral genotypes [37,38,61,62]. The predictability is further complicated by the fact that Legalon-SIL affects viral RNA replication and entry by different mechanisms [42,43,44]. Here, we confirmed the full resistance of isolate JFH1 (gt2a) against Legalon-SIL, whereas GLT1 (gt1b) RNA replication was substantially more resistant than Con1, but the infection still was efficiently inhibited by additive effects on entry. However, to this end, the model is widely limited to gt1b isolates and we will need an in-depth understanding of the determinants limiting replication of other genotypes in Huh7 cells beyond SEC14L2 to address the phenotypic analysis of therapeutically more challenging genotypes in the future. Another possible application of serum-infection assays is in the characterization of neutralizing antibody responses in vaccine development, which are so far based on collections of prototype isolates [63,64], lacking genetic intra-host diversity (reviewed in [65]).

Overall, our study established an experimental pipeline facilitating HCV infections based on patient sera, which might be helpful in developing personalized treatment regimens in hard-to-cure patients and in vaccine development.

## Figures and Tables

**Figure 1 viruses-16-01871-f001:**
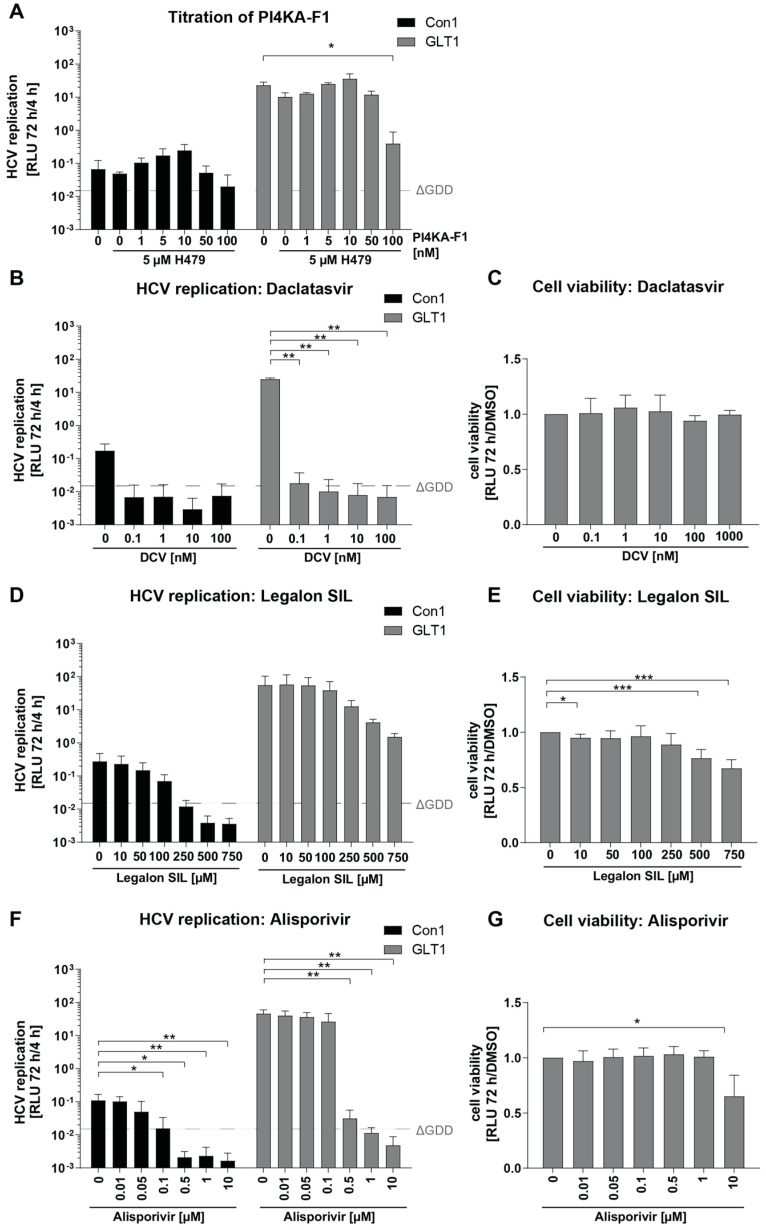
Establishment of a cell culture model to assess HCV WT replication: (**A**,**B**,**D**,**F**) Subgenomic reporter replicons based on Con1 WT and GLT1 WT were electroporated into Huh7-Lunet C/M/S cells and treated with either DMSO or the indicated concentrations of (**A**) H479 combined with PI4KA-F1 (PCi), or with PCi and the indicated concentrations of DCV (**B**)/Legalon SIL (**D**) or Alisporivir (**F**) at 4 h after electroporation. Cells were harvested 72 h later and firefly luciferase activity from cell lysates (RLU) was quantified as a correlate of RNA replication efficiency and normalized to 4 h. A replication-deficient Con1 variant (ΔGDD) served as negative control and is indicated with a dashed grey line. Values are shown as means from two (**A**,**B**) or three (**D**,**F**) independent experiments with two technical replicates each. (**C**,**E**,**G**) Huh7-Lunet C/M/S cells were seeded into 96-well plates and treated with the indicated concentrations of DCV (**C**)/Legalon SIL (**E**) or Alisporivir (**G**). After 72 h, cell viability in cell lysates was measured via Cell-titer-Glow and normalized to the DMSO control. Values are shown as means from at least three independent experiments. * *p* < 0.05, ** *p* < 0.01, *** *p* < 0.001 (unpaired two-tailed Student’s *t*-test).

**Figure 2 viruses-16-01871-f002:**
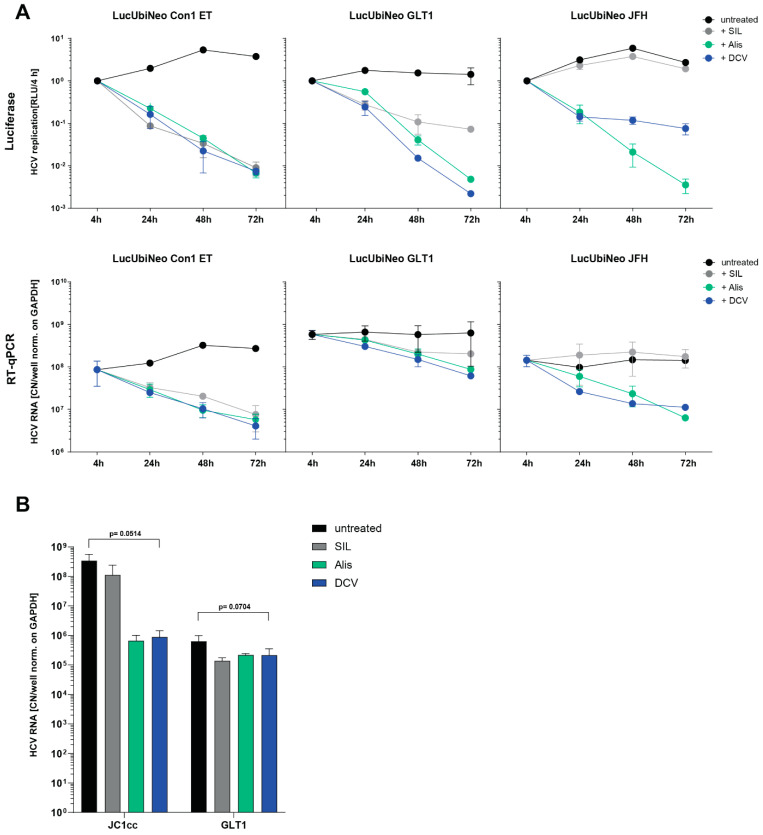
Comparison of luciferase activity and RT-qPCR-based readouts in persistent replicon cells and HCV infected cells: (**A**) Cell lines harboring the indicated replicons were treated with 250 µM SIL, 1 µM Alisporivir, 1 nM DCV, or were left untreated, and harvested at the indicated time points. Upper panels: Firefly luciferase activity from cell lysates (RLU) was quantified as a correlate of RNA replication efficiency and normalized to 4 h. Values are shown as means from two independent experiments with two technical replicates each. Lower panels: After RNA extraction, HCV RNA copies were quantified using RT-qPCR. HCV RNA copy numbers (CN) were normalized on GAPDH and are shown as CN per well. Values are shown as means from two independent experiments with three technical replicates each. (**B**) Huh7-Lunet C/M/S cells were infected with either GLT1 patient serum or JC1cc infectious particles (MOI of 10 at the first repetition, MOI of 1 at the second and third repetition). After 72 h, samples were harvested, and HCV RNA replication was measured via RT-qPCR. HCV RNA copy numbers (CN) were normalized on GAPDH and are shown as CN per well. Values are shown as means of four (GLT1: untr. + DCV), three (JC1cc: untr. + DCV), or two (GLT1 and JC1cc: SIL and Alisporivir treatment) independent experiments with three technical replicates each. *p*-values from unpaired two-tailed Student’s *t*-test are indicated for all experiments with 3 or more biological replicates.

**Figure 3 viruses-16-01871-f003:**
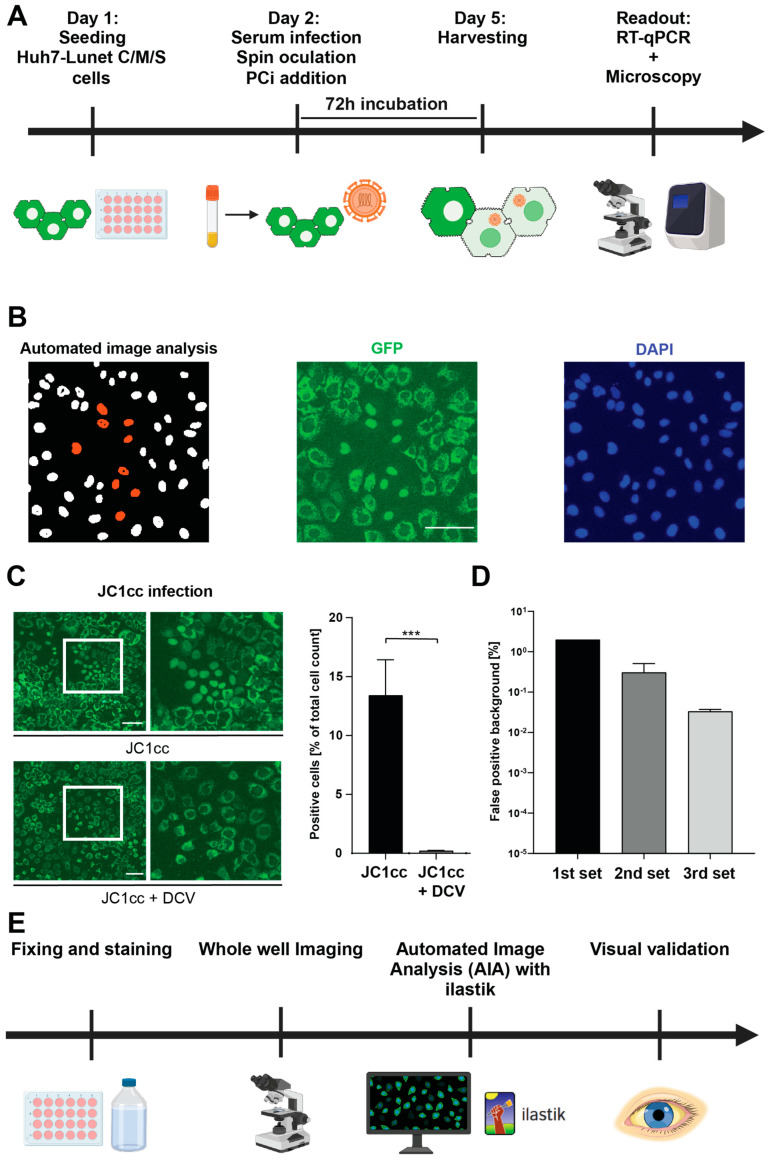
Workflow for serum infection and establishment of semi-automated image analysis: (**A**) Workflow for serum infection. (**B**) Principle of automated image analysis by machine learning. Infected cells are identified by nuclear GFP due to co-localization with DAPI. Positive cells were marked in red by ilastik, facilitating visual validation. (**C**) Huh7-Lunet C/M/S cells were infected with JC1cc (MOI = 1) alone or additionally treated with DCV, fixed, and stained with DAPI 72 h post-infection. A group of infected cells is indicated by a white box and additionally displayed with a higher magnification. Percentage of positive cells (right panel) was obtained by automated image analysis. *** *p* < 0.001 (unpaired two-tailed Student’s *t*-test). (**B**,**C**) Scale bar indicates a length of 100 µm. (**D**) Optimization of the automated image analysis. False-positive background in images of either uninfected or DCV-treated cells was reduced by iterative training of the machine learning tool, including adjustment of cell count per well and refocusing strategy. (**E**) Workflow of automated image analysis (AIA) followed by visual validation.

**Figure 4 viruses-16-01871-f004:**
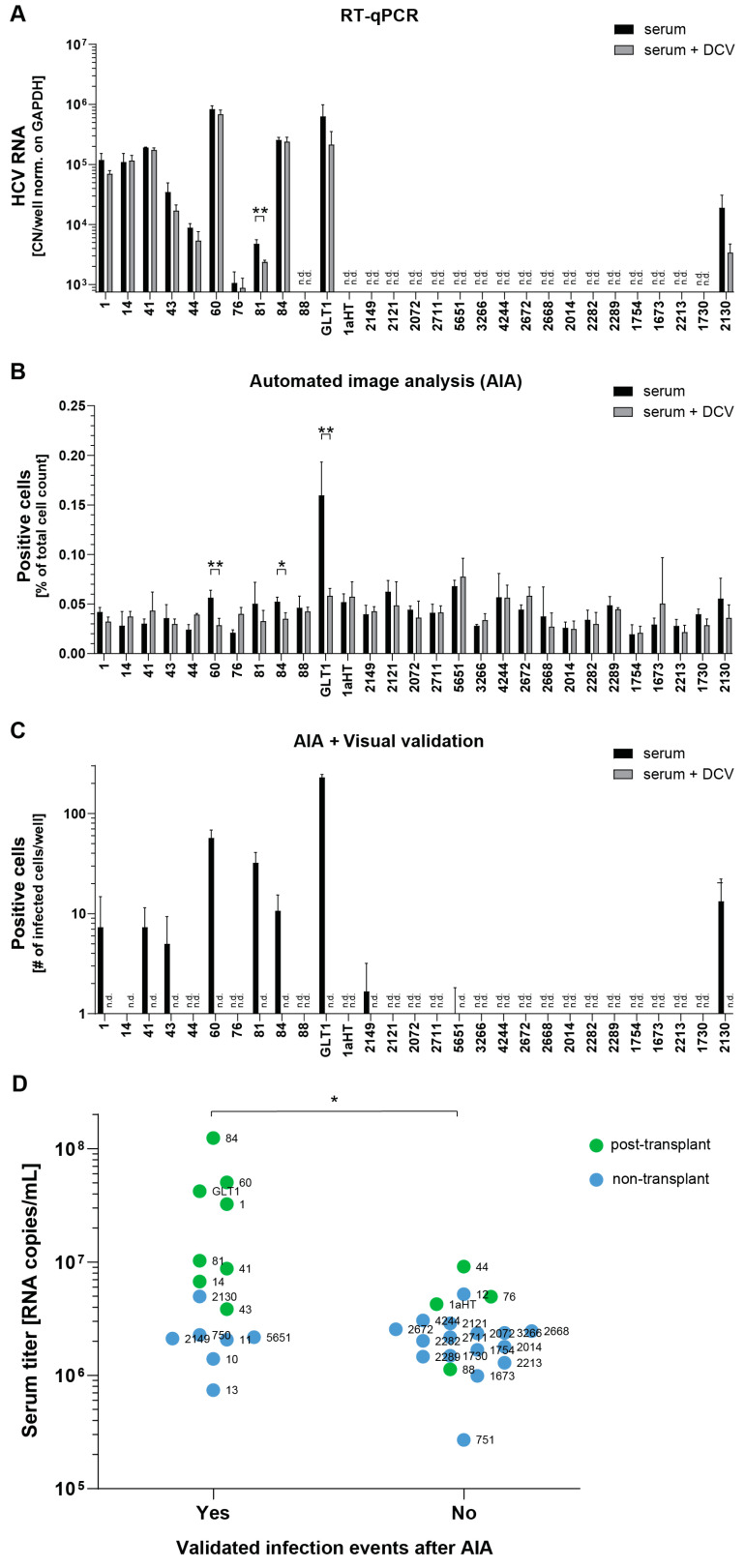
Comparison of HCV infection using high-titer sera by RT-qPCR or microscopic analysis: (**A**–**C**) Huh7-Lunet C/M/S cells treated with PCi and infected with patient sera in presence or absence of DCV; 72 h after infection, cells were fixed, stained with DAPI. (**A**) Total RNA was extracted 72 h after infection and HCV RNA was quantified. (**B**) 72 h after infection, quantification by automated image analysis was performed. (**C**) Truly infected cells detected after visual validation of hits generated by the automated image analysis. Note that no bona fide infected cells were detected in the DCV-treated samples. All values are means and SD of three or two (serum 44) independent biological replicates. Note that RT-qPCR data from the GLT1 serum were taken from Figure 2B to allow a direct comparison (n = 4). n.d.: not detectable. * *p* < 0.05, ** *p* < 0.01 (unpaired two-tailed Student’s *t*-test). (**D**) Results of the semi-automated image-based analysis of all data shown in panel (**C**) and in Figure 6C were analyzed regarding the impact of RNA titer and post-transplant status on infection outcome. The anonymized abbreviation of each serum is given next to the respective dot (see Table 1 and Table 2). Samples were judged positive if at least one infected cell was found via the automated image analysis combined with visual validation. * *p* < 0.05 (unpaired two-tailed Student’s *t*-test).

**Figure 5 viruses-16-01871-f005:**
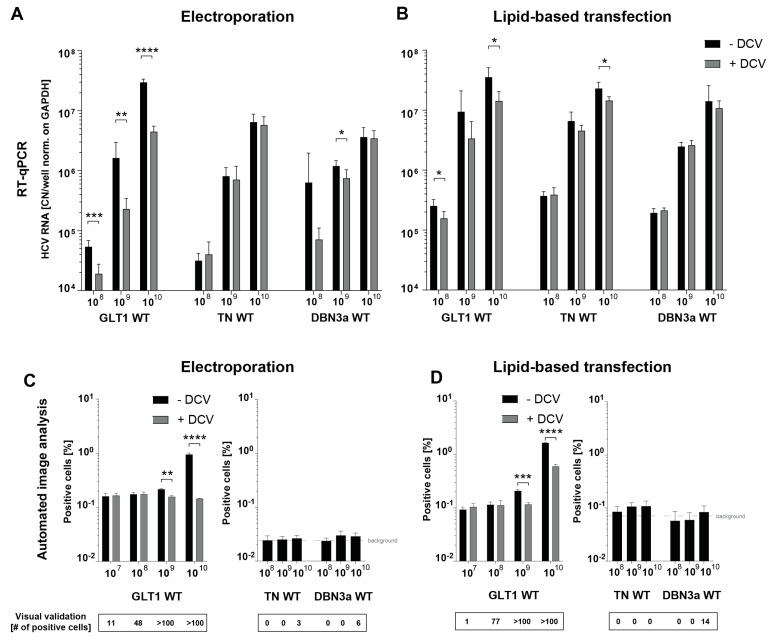
Establishment of an HCV WT RNA transfection protocol. In vitro transcribed RNA of GLT1 WT (gt1b), TN WT (gt1a), and DBN WT (gt3a) was transfected into Huh7-Lunet C/M/S cells with different amounts of input RNA, ranging from 10^7^ to 10^10^ RNA copies per transfection, either by electroporation (**A**,**C**) or by lipid-based transfection (**B**,**D**). RT-qPCR (**A**,**B**) and the automated image analysis served as readouts (**C**,**D**). Samples were measured in technical triplicates. In the image-based analysis (**C**,**D**), the samples transfected with 10^8^ RNA copies of TN WT and DBN WT were set as false-positive backgrounds, since no infected cells could be detected upon visual validation, depicted as a dashed grey line. Numbers of cells visually validated represent only DCV- samples. (**A**,**C**) Electroporation: Samples were harvested after 72 h. Values are taken from two (TN WT, DBN WT) or three (GLT1 WT) independent experiments. (**B**,**D**) Lipid-based transfection: Cells were split 24 h post-transfection, at this time point DCV was added. Samples were harvested after 2 (to stick to the total incubation time of 72 h) or 3 (to ensure DCV treatment for 72 h) days. Since there were no relevant differences in the results with varying incubation times observed, the data of the two experiments are shown in the same graph. * *p* < 0.05, ** *p* < 0.01, *** *p* < 0.001, **** *p* < 0.0001 (unpaired two-tailed Student’s *t*-test).

**Figure 6 viruses-16-01871-f006:**
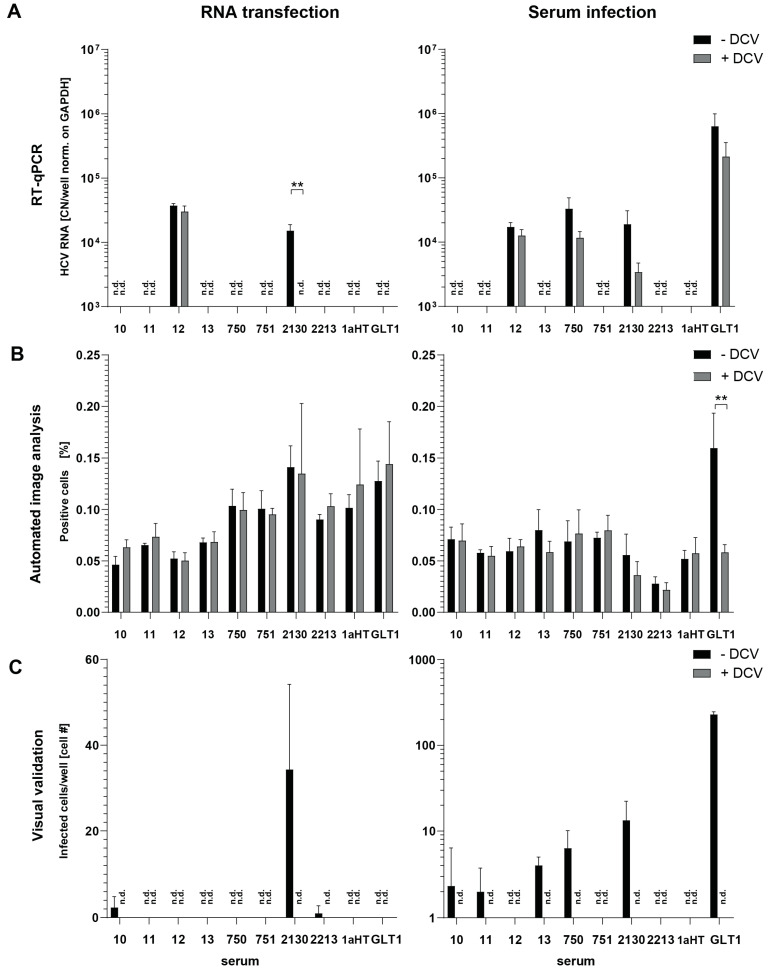
**Results of RNA transfection in comparison to serum infection**. Ten patient sera were tested using RNA transfection and serum infection in parallel. All samples were harvested after 72 h, treatment with DCV served as negative control. Samples were analyzed via RT-qPCR (**A**), AIA (**B**), and with AIA followed by visual validation (**C**). Values are shown as means of three biological replicates and were measured in technical triplicates in RT-qPCR. Data of the serum infection assay of sera 2130, 2213, AU, and GLT1 were taken from Figure 4 for better comparison. n.d.: not detectable. ** *p* < 0.01 (unpaired two-tailed Student’s *t*-test).

**Table 1 viruses-16-01871-t001:** Sera used for comparative analysis of RT-qPCR and Automated Image Analysis (AIA).

Serum	Date and Cohort	Titer ^1^	LTx ^2^ y/n	Gt	HCV RNA pi ^3^	Infected Cells ^4^	Difference to DCV Control *p* < 0.05 (RT-qPCR/AIA)
1	2010 Barcelona	3.24 × 10^7^	y	1b	Yes	Yes	No/No
14	2011 Barcelona	6.74 × 10^6^	y	1b	Yes	Yes	No/No
41	2012 Barcelona	8.74 × 10^6^	y	1b	Yes	Yes	No/No
43	2012 Barcelona	3.84 × 10^6^	y	1b	Yes	Yes	No/No
44	2012 Barcelona	9.11 × 10^6^	y	1b	Yes	No	No/No
60	2012 Barcelona	5.04 × 10^7^	y	1b	Yes	Yes	No/Yes
76	2013 Barcelona	4.95 × 10^6^	y	1b	Yes	No	No/No
81	2014 Barcelona	1.03 × 10^7^	y	1b	Yes	Yes	Yes/No
84	2014 Barcelona	1.24 × 10^8^	y	1b	Yes	Yes	No/Yes
88	2014 Barcelona	1.13 × 10^6^	y	1b	No	No	No/No
1aHT	2016 Heidelberg	4.25 × 10^6^	y	1a	No	No	No/No
GLT1	2014 Heidelberg	4.22 × 10^7^	y	1b	Yes	Yes	No/Yes
2149	2015 Frankfurt	2.11 × 10^6^	n	1b	No	Yes	No/No
2121	2015 Frankfurt	2.87 × 10^6^	n	1b	No	No	No/No
2072	2015 Frankfurt	2.35 × 10^6^	n	1b	No	No	No/No
2711	2015 Frankfurt	2.17 × 10^6^	n	1b	No	No	No/No
5651	2016 Frankfurt	2.17 × 10^6^	n	1b	No	Yes	No/No
3266	2015 Frankfurt	2.37 × 10^6^	n	1b	No	No	No/No
4244	2015 Frankfurt	3.06 × 10^6^	n	1b	No	No	No/No
2672	2015 Frankfurt	2.55 × 10^6^	n	1b	No	No	No/No
2668	2015 Frankfurt	2.46 × 10^6^	n	1b	No	No	No/No
2014	2015 Frankfurt	1.79 × 10^6^	n	1b	No	No	No/No
2282	2015 Frankfurt	2.02 × 10^6^	n	1b	No	No	No/No
2289	2015 Frankfurt	1.46 × 10^6^	n	1b	No	No	No/No
1754	2015 Frankfurt	1.68 × 10^6^	n	1b	No	No	No/No
1673	2015 Frankfurt	9.88 × 10^5^	n	1b	No	No	No/No
2213	2015 Frankfurt	1.29 × 10^6^	n	1b	No	No	No/No
1730	2015 Frankfurt	1.49 × 10^6^	n	1b	No	No	No/No
2130	2015 Frankfurt	4.97 × 10^6^	n	1b	Yes	Yes	No/No

^1^ [RNA copies/mL]; ^2^ Serum post LTx; ^3^ HCV RNA detected post-infection (pi) with a Ct < 36; ^4^ Automated image analysis (AIA) verified by visual control.

**Table 2 viruses-16-01871-t002:** Patient Sera used for electroporation (epo) after RNA extraction. Sera 2130, 2213, 1aHT, and GLT1 were already used before; thus, they are also described in Table 1.

Serum	Date of Acquisition	Serum Titer ^1^	Gt	LTx ^2^ y/n	Serum Volume	Input Infection ^3^	Input epo ^3^
10	2015 Freiburg	1.40 × 10^6^	1b	n	10 mL*	1.12 × 10^5^	9.06 × 10^5^
11	2015 Freiburg	2.07 × 10^6^	1b	n	10 mL*	1.66 × 10^5^	4.89 × 10^5^
12	2015 Freiburg	5.20 × 10^6^	1b	n	10 mL*	4.16 × 10^5^	5.67 × 10^6^
13	2015 Freiburg	7.41 × 10^5^	1b	n	10 mL*	5.93 × 10^4^	1.02 × 10^6^
750	2022 Heidelberg	2.28 × 10^6^	1b	n	10 mL*	1.82 × 10^5^	2.70 × 10^4^
751	2022 Heidelberg	2.69 × 10^5^	1b	n	10 mL*	2.15 × 10^4^	2.10 × 10^4^
2130	2015 Frankfurt	4.97 × 10^6^	1b	n	1 mL	3.98 × 10^5^	1.06 × 10^6^
2213	2015 Frankfurt	1.29 × 10^6^	1b	n	1 mL	1.03 × 10^5^	7.31 × 10^5^
1aHT	2016 Heidelberg	4.25 × 10^6^	1a	y	1 mL	3.40 × 10^5^	6.19 × 10^4^
GLT1	2014 Heidelberg	4.22 × 10^7^	1b	y	1 mL	3.38 × 10^6^	1.22 × 10^6^

^1^ [RNA copies/mL]; ^2^ Serum post LTx; ^3^ [RNA copies] per well; * known storage history, no freeze/thaw cycles.

## Data Availability

The original contributions presented in this study are included in the article/Appendix A. Further inquiries can be directed to the corresponding author(s).

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
