# Peer review of "Detection of Hepatitis C Virus Infection from Patient Sera in Cell Culture Using Semi-Automated Image Analysis"

_viruses, 2024, doi:10.3390/v16121871_

Round 1

Reviewer 1 Report

Comments and Suggestions for Authors

Hepatitis C virus (HCV) is a major global pathogen, leading to liver cirrhosis and hepatocellular carcinoma. In this manuscript, Schäfer et al. constructed Huh7-Lunet C/M/S cells and investigated the effects of the PI4KA inhibitor, Legalon SIL, and the direct-acting antiviral agent, Declatasvir, on HCV replication across several genotypes. Additionally, the authors tested sera, primarily from HCV genotype 1b-infected patients, to assess their capacity to establish successful RNA replication in Huh7-Lunet C/M/S cells. They also developed a novel quantitative, image-based analysis method with visual validation that is capable of detecting even single infection events. This paper provides valuable insights for researchers and clinicians in the field of HCV. While the paper is well-written and suitable for publication, several points should be addressed before final acceptance.

Major

1.       In Figure 1A, treatment with 5 nM PI4KA-F1 enhances HCV replication compared to the 0 nM PI4KA-F1 condition in the presence of 5 µM H479. However, this effect does not appear to be significantly more pronounced when compared to the untreated H479 condition, likely because H479 treatment itself reduces HCV replication. This suggests that the use of PI4KA-F1 may not have a substantial impact on HCV replication under these conditions. The authors should address why the effect of PI4KA-F1 is relatively modest in this context and discuss its relevance in comparison to the untreated control.

2.       As mentioned in lines 343-346, I agree with the suggestion that Legalon SIL may work through a multistep mechanism to suppress GLT1 HCVcc infection, which could explain why its effect is observed in this context, unlike in RNA electroporation or cell lines harboring replicons. However, in Figure 1D, Legalon SIL does not reduce luciferase expression from the GLT1 subgenomic replicon below its delta GDD, even up to 750 µM. The authors should discuss why the effect of SIL is partial or absent in GLT1 (1b) and JFH1 (2a) compared to Con1 (1a), and how this may depend on the differences in genotypes.

3.       In Figure 4C, upon long-term culture of the positive clones, did you observe cells that continuously replicate HCV RNA or produce infectious particles?

4.       In Figure 5, the lack of difference between DCV treatment and no treatment in the RNA electroporation of TN (1b) and DNB3a (3a) suggests that HCV RNA replication does not occur.  This appears to imply that the pan-genotypic effect of SEC14L2 and PCi may be absent. Have the authors confirmed the protein expression of SEC14L2 in Huh7 Lunet C/M/S cells and the phosphorylation level of PI4KA upon PCi treatment through Western blot analysis?

5.       In Line 649, the authors mention testing HCV RNA electroporation and lipofection to avoid factors interfering with serum infection. The authors should consider removing neutralizing antibodies from the serum to assess whether this results in increased infectivity. It would be very useful to investigate whether removing antibodies improves infectivity in CHC patients who have not undergone liver transplantation, as this could potentially show a different effect compared to liver transplantation recipients, where no significant change in infectivity is observed.

Minor point

1.       If there are any specific criteria for visual evaluation, please include them in the Materials and Methods section.

2.       In Table 3, is "RNA pi" an abbreviation for RNA post-infection?

3.       Table 1 follows Table 3, please correct the table numbering

4.        In Line 540, the font size of (B, D) is bigger than the others . Please adjust the font size.

Reviewer 2 Report

Comments and Suggestions for Authors

Dear Authors.

The manuscript entitled “Detection of hepatitis C virus infection from patient sera in cell culture using semi-automated image analysis” aims to assess and optimise a serum derived infection assay using a semi-automated image analysis.  The authors use a renilla assay and qRT-PCR to validate improved infection events using ectopically expressed SEC14L and titrated PCi as previously published. They then aim optimise the assay using patients isolates by establishing an automated image analysis pipeline incorporating a machine learning tool.  The research question is important and well justified in the field of HCV infection as this tool maybe useful for measuring patient specific resistance to DAAs or characterisation neutralising antibody responses in vaccine development.  The research contributes well to the previously published work on SEC14L and PCi in the aim to improve the detection of patient sera derived infection in cell culture. However, there are several points that could be modified that would improve the quality of the manuscript and these are listed below:

Minor points:

1.       line 137 full names of abbreviated drugs are found later in the text (line 171-2) and should be cited here

2.       Line 183 (and or line 117) JC1cc should be explained in more detail (i.e. genotype, chimera of etc)

3.       line 190 methods or reference to method should be added for the TCID50 assay and calculation.

4.       Line 201 “JFH1 based plasmids” needs to be clearer as JFH1 is not mentioned anywhere else in the manuscript.

5.       Line 204 abbreviation “CN” should be defined

6.       Line 228-234 More details on the Electroporation method (or reference to a detailed methodology) is needed.

7.       Line 248 define TNE buffer

8.       Lines 146-167, how many images or how many cells were captured/counted per condition?

9.       Table 3, Pi abbreviation should be defined, table 3 should be labelled, table 1 and table 1 should be labelled table 2 and table 2 should be labelled table 3.

10.   The order of serum numbers of in table 1 and figure 4 should be adjusted so they are in the same order.

Major points:

1.       Figure 2 and Lines 327-351.  

a.       The results observed in figure 2b do not entirely reciprocate what is written in the text, for example, there is a ½ log decrease in HCV RNA observed in figure 2B when comparing untreated and SIL treated JC1cc where as the text reads “Jc1cc RNA levels were …”not affected by Legalon-SIL”

b.       Why are there statistics for only untreated versus DCV? 

c.       In the text it is written “In the case of GLT1 serum infection, no pronounce additive effect of the combination of PCi treatment and SEC14L2 overexpression in comparison to PCi treatment alone could be detected when compared to the earlier results” this comment is misleading as no comparison is made here. Alternatively this experiment could be improved with additional controls of +/- H479 and PCi

d.       These experiments are done mostly with 2 biological replicates and a third biological replicate for both “A and “B would improve the result.

e.       Fig2a could be simplified and statistics included if only the 72h timepoint is shown as with figure 1A and 2B.

2.       Table 3, could be deleted and the results from table “1” could be used for both rtQPCR and image analysis to reduce bias. Could all the samples from Table 2 also be used in the table 1 analysis for completeness.

3.       Figure 3, image scales are needed, Fig 3A and 3B could be combined to compare the 2 analyses including timeframe for both., Fig 3D can be deleted, the text is enough, fig 3C a zoomed in version of Jc1cc (like the top right panel) vs Jc1cc DCV would be more appropriate.

4.       Lines 613-621, the number of sera analysed and detectable infection events are puzzling (15/34) and should be clarified. For example, there are 27 patient sera (2 positive RNA post infection) in table 3, a further different 25 patient sera (10 infection events) in table 1 and 6 other patient sera (of which 4 were positive after serum infection) in table 2.   This also puzzling in the number of transplanted versus non transplanted (8/11 vs 7/24 =15/35), if theses results were combined in 1 table it would be much easier to follow.

5.       Line 632- it would be interesting to sequence some of the positive infection sera post infection to see if there are changes similar to HCVcc adaptive mutations.

6.       Line 663- it would be interesting to check more clinically relevant genotypes/subtypes in terms of drug resistance i.e., Gt4 (sub-saharan isolates).

Round 2

Reviewer 2 Report

Comments and Suggestions for Authors

Dear Authors

I am satisfied with the changes made to the manuscript, I would just add one minor edit for completeness:

Line 120 add in brackets, (HCVcc) so that further reference to HCVcc is complete.

Author Response

Comment: Line 120 add in brackets, (HCVcc) so that further reference to HCVcc is complete.

Response: We thank the reviwer for his input and added the missing information in line 120.